# Review of Thermoelectric Generators at Low Operating Temperatures: Working Principles and Materials

**DOI:** 10.3390/mi12070734

**Published:** 2021-06-22

**Authors:** Nurkhaizan Zulkepli, Jumril Yunas, Mohd Ambri Mohamed, Azrul Azlan Hamzah

**Affiliations:** 1Institute of Microengineering and Nanoelectronic (IMEN), Universiti Kebangsaan Malaysia (UKM), Bangi 46300, Malaysia; khaizan2821@uitm.edu.my (N.Z.); ambri@ukm.edu.my (M.A.M.); 2Centre of Foundation Studies, Universiti Teknologi MARA, Cawangan Selangor, Kampus Dengkil, Dengkil 43800, Malaysia

**Keywords:** energy harvester, low operating temperatures, thermoelectric generator, thermoelectric material

## Abstract

Thermoelectric generators (TEGs) are a form of energy harvester and eco-friendly power generation system that directly transform thermal energy into electrical energy. The thermoelectric (TE) method of energy harvesting takes advantage of the Seebeck effect, which offers a simple solution for fulfilling the power-supply demand in almost every electronics system. A high-temperature condition is commonly essential in the working mechanism of the TE device, which unfortunately limits the potential implementation of the device. This paper presents an in-depth analysis of TEGs at low operating temperature. The review starts with an extensive description of their fundamental working principles, structure, physical properties, and the figure of merit (ZT). An overview of the associated key challenges in optimising ZT value according to the physical properties is discussed, including the state of the art of the advanced approaches in ZT optimisation. Finally, this manuscript summarises the research status of Bi_2_Te_3_-based semiconductors and other compound materials as potential materials for TE generators working at low operating temperatures. The improved TE materials suggest that TE power-generation technology is essential for sustainable power generation at near-room temperature to satisfy the requirement for reliable energy supplies in low-power electrical/electronics systems.

## 1. Introduction

The pursuit of new renewable energy options, as well as preserving a sustainable energy system, have become a global priority. Energy harvesting is the process of harnessing energy from available resources and subsequently converting it to electricity [1]. In the absence of the main energy source, harvested energy is typically used to operate small and medium-sized computer devices, as well as electrical systems, with power production varying from nW to hundreds of mW [2,3]. It must be considered that the main aim of energy harvesting is not to produce power at a large scale, but to capture the wasted energy, put in storage, and use it later for the daily operation of micro- and nanoelectronic systems [4,5,6].

Examples of energy harvesters’ concepts are based on the photoelectric effect [7] and electromagnetic [8,9], piezoelectric [10,11,12], and radio-frequency harvester mechanisms [13,14]. Among all energy-harvester devices, TEGs have received extensive attention due to their ability to interconvert heat and electricity [15], as well as their large potential energy availability [16,17] and versatile application [18,19]. This allows TE to play a prominent role as a renewable energy source by repurposing excess heat from the environment [20,21].

A TE energy harvester, which can generate electricity, has a lot of advantages, including the following: usage as an extremely durable and lightweight solid-state system, while being maintenance-free, silent, and environmentally friendly, since heat is harvested from normally wasted heat sources and converted to electricity; a high-temperature process (up to 250 °C); and scalable applications that are capable of harvesting large quantities of energy when required. Additionally, power can be harvested from hot or cold surfaces [22]. Furthermore, a TEG unit generates usable renewable electricity without consuming fossil fuels, resulting in a decrease in greenhouse gas pollution and therefore ecofriendly. Due to these numerous advantages, TE technology remains highly competitive.

TE phenomena are exhibited in almost all conducting materials. Since the figure of merit is temperature-dependent, a more precise measure of performance is the dimensionless figure of merit ZT, where T is the absolute temperature. Only materials with a 0.5 ZT are commonly considered as TE materials [23]. To produce sufficient power, a TE generator must be efficiently used across a large temperature difference ΔT = T_h_ − T_c_ (T_h_ = temperature of hot side, T_c_ = temperature of cold side). Working temperature of the TE material depends on the material that has the highest ZT value [24]. Table 1 shows the TEG working temperature ranges and variety of waste-heat sources.

The ZT values of some common commercially used materials are shown in Figure 1. These materials can be divided into three groups based on the working temperature range:(1)Low-temperature materials, in which the working temperature is around 400 K. Alloys based on bismuth are among the popular materials in this temperature region [40].(2)Intermediate-temperature materials, in which the working temperature is between 600 and 900 K. Alloys based on lead are commonly used in this category.(3)High-temperature materials, in which the working temperature is higher than 900 K. Material fabricated from silicon–germanium usually falls under this region [21].

An example of the implementation of TE devices is in powering the spacecrafts of most NASA deep-space missions in the form of a radioisotope thermoelectric generator, exploiting the high-temperature gradient in the system [42]. The potential applications of TE in the low-operating-temperature range have been widely described in the literature. The development of new techniques has made it possible for TE to become an alternative power supply for various low-power devices. A variety of applications of low-operating-temperature TEGs are summarized in Table 2.

The numerous applications of low-operating-temperature TEGs proves that they play an important role as an alternative energy source. Nonetheless, the TE performance of such materials is relatively low in the low-temperature region (300–400 K). In this low-temperature region, the ZT_avg_ value of TE materials is roughly 1, and the theoretical efficiency is only around 5.6% [30]. As a result, there is a need for high-performance TE devices running at low temperatures, which are critical in recycling low-grade heat waste.

On the other hand, it was revealed that almost 72% of primary energy used globally is wasted in the form of excess heat [56]. According to the temperature and dispersion of waste heat (Figure 2), low-grade wasted heat with a temperature of less than 200 K accounts for approximately 63%, mostly from power, transportation, and industrial waste [57]. As a result, there is a need to improve TE efficiency in the low-temperature range.

In this paper, we summarise the analysis of the working principle of TEG devices and their materials, focusing on TEG workings at low operating temperatures. In this review, the primary working mechanisms of TEGs and their material properties are introduced. Then, the paper emphasises the key challenges associated with the ZT value according to their physical properties while highlighting the state-of-the-art advanced approaches in ZT optimisation. Then, we summarise the performance of Bi_2_Te_3_-based semiconductors and other compound materials in recent years. Finally, the review points out some of the challenges for future work to improve TE device performance.

## 2. Working Mechanisms of Thermoelectric Generators

### 2.1. The Basic Principle and Structure

In 1821, German physicist Thomas Johann Seebeck discovered heat conversion into electricity when two different metals at different temperatures were applied between junctions. This phenomenon is known as the Seebeck effect. Today, TEGs apply the Seebeck effect to convert heat into electricity [58].

The basic principle of a TEG is the production of electromotive force (EMF) across a material when temperature differences are available. Here, the voltage difference produced between the two metal junctions is proportional to the temperature difference.

The TEG is a solid-state heat engine made of two semiconductor materials with primary junctions, known as the *n*-type and *p*-type. The *n*-type elements are doped to contain a highly concentrated negative charge or electron, while the *p*-type element is doped to make it into a high-quantity positive charge, or holes. The elements with a high number of electrons will provide a negative Seebeck coefficient. On the other hand, elements with a high number of holes contribute to a positive Seebeck coefficient [59].

A TEG is basically made up of one or more TE couples. The most basic TEG is a thermocouple, which comprises a pair of *p*- and *n*-type TE materials, electrically bound in series and thermally bound in parallel. An electrical conductor, normally a copper strip, connects the two legs to one side, creating a junction [24]. As seen in Figure 3, the *p*- and *n*-type materials are located between the cold and hot side to provide a temperature-difference condition.

As the temperature between the junction changes, the electron at the end of the *n*-type material closest to the heat source gains energy. At the end of the *n*-type element closest to the heat source, the electron concentration is greater than at the end of the cold side. The electron at the heat source end will move to the cold side end, similar to how heat travels from a hot region to a cold region [60]. The same applies to the holes in the *p*-type elements.

When a potential load exists, the movement of electrons and holes in an *n*- or *p*-type element induces the EMF, which results in electrical current production. As long as a temperature gradient between the two junctions exists, current will be produced. More electric output power is produced as the temperature gradient, ΔT = T_h_ − T_c_, across the TEG unit rises.

The formation of EMF in a pair of *n*- and *p*-type elements is minimal. Consequently, many such TE couples are needed to obtain a high enough voltage. Several TE couples are electrically linked in series and sandwiched between two ceramic sheets [61].

A TEG is a heat engine [23]. Therefore, efficiency is the most influential parameter for indicating device performance. The efficiency of the TEG is described as the ratio of the electrical power supplied to the load versus the heat absorbed at the hot junction if the converter is used as an ideal generator with no energy loss. For a TEG operating between a heat source at high temperature, T_h_, and a heat sink at low temperature, T_c_, the TEG efficiency is given by [62]:
(1)$$\eta_{TEC}=\frac{T_{h}-T_{c}}{T_{h}}\frac{\sqrt{\left(1+ZT\right)}-1}{\sqrt{1+ZT}+T_{h}/T_{c}}$$

As can be seen in Equation (1), the efficiency of TEG is highly dependent on the temperature gradient between the junctions, the TEG operating temperatures, and the ZT value. The$\;\eta_{TEG}$ proportionally increases as the temperature gradient between the junctions, ΔT, increases. For TEGs at low operating temperatures, 25% of the $\eta_{TEG}$ can be reached if only materials with ZT = 10 are obtainable [41].

### 2.2. Physical Properties of Thermoelectric Materials

At the beginning of the 20th century, the advancement of TE materials began. Altenkirch suggested a dimensionless figure of the ZT value, called the figure of merit, to indicate the TE material’s efficiency:(2)$$\mathrm{ZT}=\frac{S^{2}\sigma T}{\kappa }$$
where S is the Seebeck coefficient determined by the ratio of the EMF produced to the temperature difference. In addition, σ is the electrical conductivity in S/cm, κ is the thermal conductivity (including lattice component k_p_ and electronic component k_e_) in W/mK, and T is the absolute temperature in K [62].

The Seebeck effect is the EMF produced across a material in which temperature differences are present. The Seebeck effect, S, is the ratio of the EMF produced to the temperature difference. The S of an intrinsic semiconductor (undoped semiconductor) is determined by the partial S of the electrons and holes, weighted by their conductivities, according to the relation:(3)$$S=\frac{S_{e}\sigma_{e}+S_{h}\sigma_{h}}{\sigma_{e}+\sigma_{h}}$$
where S_e_ is the Seebeck component due to *n*-type carrier conduction, S_h_ is the Seebeck component due to *p*-type carrier conduction,$\;\sigma_{e}$ is the electrical conductivity by *n*-type carriers, and $\sigma_{h}$ is the electrical conductivity by *p*-type carriers.

Here, ZT is a material property. Enhancing the ZT value will improve the efficiency of a TE material. In order to obtain it, the numerator in Equation (2), S^2^σ, which is also called a “power factor” (PF), needs to be maximised. Other than that, the denominator, κ, should be held as low as possible [63]. In the ZT equation, T is the temperature in Kelvin where all the property is being measured at that particular temperature, with small changes applied to determine Z.

The fundamental criteria for a good TE material in a semiconductor are that it exhibits both high electronic properties and very low κ values. The concept of a “phonon-glass electron-crystal” (PGEC) to describe the properties of desired TE material was first introduced by Slack [64]. This ideal material posses a physical property that would scatter phonons, reducing the lattice contribution to κ as if in glass, but at the same time allowing electrons to move easily through the material, as they would in a crystal. The term ‘phonon’ refers to heat. Heat is transferred in solids through atomic lattice vibrations. Chemical bonds hold the atoms in solids together in a lattice configuration. These bonds are not rigid; rather, they behave like a spring, connecting the atoms through a spring–mass mechanism. As an atom or a plane of atoms displaces, the displacement will spread through the crystal as a wave, carrying energy with it. These waves are quantised and are known as phonons [65].

## 3. Challenges in Optimisation of ZT Value

In the following sections, the challenges posed by the physical properties of TE to achieve a PGEC material are explained, and we distill rational design strategies to achieve materials with high TE efficiency. As based on Equation (2), the improvements of PF and κ are the two important approaches that determine the ZT value.

### 3.1. Challenges in the Optimisation of PF

To obtain an improved ZT value, the PF as the numerator in Equation (3) needs to be maximised. This is related to an improvement in the material properties, such as S and σ, which rely on the transport parameter, while the transport parameters are linked to charge characteristics that involve the concentration of the carrier, the mobility of the carrier, and the effective mass ($m^{*}$) of the carrier.

To ensure that S is large, there should only be a single type of carrier. Mixed *n*-type and *p*-type conduction will lead both charge carriers to move to the cold end, cancelling out the induced Seebeck voltages. Among the biggest strengths offered by semiconductors is the ability to adjust electrical conductance through doping. Low carrier concentration (*n*) can contribute to greater S, as can be seen in Equation (4), based on the electron-transport model:(4)$$S_{e}=\frac{8\Pi^{2}k_{B}^{2}}{3eh^{2}}m^{*}T{\left(\frac{\Pi }{3n}\right)}^{\frac{2}{3}}$$
where *n* is the carrier concentration, and $m^{*}$ is the effective mass of the carrier. However, according to Equation (5), σ is directly proportional to *n*:(5)σ=neμ
where *μ* is carrier mobility. As observed in Equations (4) and (5), σ is low for low-*n* material, making it difficult for the material to achieve a high PF.

*μ* determines how fast a carrier, electron, or hole can move in a solid material [66]. $m^{*}$ on the other side corresponds to the density of effective state mass at the surface of Fermi. *μ* and$\;m^{*}$ are inversely proportional [67]. Higher$\;m^{*}$ carriers will travel at slower speeds. As a result of the poor mobilities, Equation (5) projects that σ is also low. This relation of S, σ, and *μ* is known as the Pisarenko relation [21]. This characteristic is another challenge in improving the PF. A large $m^{*}$ results in a strong S (see Equation (4)), but poor σ and *μ*.

### 3.2. Challenges in Reducing the Thermal Conductivity, κ

Another challenge in improving the ZT value is the minimisation of the thermal conductivity, κ. The κ reduction is very complicated, given that it must occur without hampering the electronic properties. κ is governed by two primary components: electronic thermal conductivity,k_e_, and lattice thermal conductivity, k_p_. The relationship between the κ component and electronic properties, along with the strategies to minimise them, are explained in the following sections.

#### 3.2.1. Electronic Thermal Conductivity, k_e_

k_e_ represents the charge carriers’ contribution and is directly related to σ through the Wiedemann–Franz law, as described in Equation (5):(6)$$k_{e}=L_{O}\sigma T$$
where L is the Lorenz factor, which varies between 1.5 to 2.44 × 10^−8^ WΩ K^−2^ in the semiconductor, σ; and T is the absolute temperature [68,69]. From Equation (5), it can be observed that k_e_ is proportional to σ. It is a challenge to minimise κ. Therefore, the challenge to minimize κ is the small k_e_, which also corresponds to low PF.

#### 3.2.2. Lattice Thermal Conductivity, k_p_

k_p_ is caused by the vibration of atoms relative to their equilibrium location in a solid material. These are referred to as phonons, and they spread via solids through wave packets in the presence of a temperature gradient.

Crystalline materials are good TE candidates due to their ability to scatter phonons without significantly disrupting σ. The heat flow is carried by mean free paths from less than 10 nm to greater than 1 μm, creating a need for phonon-scattering agents at a variety of length scales [40]. In a conventional crystalline solid, k_p_ is directly related to mean free paths, as described in Equation (6):(7)$$k_{P}=\frac{1}{3}{\mathrm{CV}}_{S}l$$
where C is the specific heat capacity, V_S_ is the velocity of sound, and l is the mean free paths [69]. From Equation (6), it can be observed that k_p_ is proportional to mean free paths. Low mean free paths result in a low k_p_, which is also correlated to a high ZT.

The traditional ways of achieving low k_p_, such as forming solid solutions or making composite structures, typically have a detrimental effect on *μ*. Much of the research that has gone into modern TE materials began with theoretical calculations on quantum well superlattice structures by Dresselhaus [70] in the early 1990s. This initial research showed that if layers of Bi_2_Te_3_ could be made with 1 nm thickness, the ZT of the Bi_2_Te_3_ superlattice structures could be near 13. This is possible since electrons would be restricted to move in only two dimensions, with the additional benefit of scattering phonons as they move between layers. If the crystals can be reduced to nanoscale lengths, then the number of interfaces could be high enough to impede phonons from moving freely. This work by Dresselhaus shed light on new directions in the field of thermoelectricity, focusing on the role of nanostructures.

## 4. Advanced Approaches in Optimisation of ZT Value

### 4.1. Advanced Approaches in PF Optimisation

Attempts to optimise the PF of TE material have been reported by some researchers. The difficulty in optimising the PF is that *n*, carrier *μ*, and ${\text{m}}^{*}$ are all highly dependent on one another. Herein, a summary of advanced approaches in PF optimisation has been given. Different advanced approaches to enhance PF are shown in Table 3.

Enhancing $m^{*}$ leads to a high S, which will subsequently contribute to PF optimisation. Among the methods to increase $m^{*}$ is band engineering. This approach was successfully exploited by She et al. [71]. They reported that the increase of $m^{*}$ contributed to the enhanced band $m^{*}$ through Cu additives in BiSe. This same approach of band engineering was adopted by Tan et al. [74] by alloying CdTe in *n*-type PbTe, resulting in a PF 3 times higher. Another technique utilising a similar concept is bandgap convergance [72,73]. Adding selenium (Se) to the TE material at a medium operating temperature raised the PF value. Du et al. used a single parabolic band model to calculate $m^{*}$, and found that Se substitution expanded the narrow bandgap and reduced the Ge vacancy, resulting in improved electrical transport properties. On the other hand, according to He et al. Se alloying in SnS promoted the interplay of the three valence bands in SnS, responsible for the optimization between $m^{*}$ and *μ*.

In order to further enhance PF value, *μ*, which determines how fast a carrier can move in a solid material, is another material property that needs to be increased. Several methods to improve *μ* have been reported in the literature. Modulation doping is a method that is frequently used to increase *μ*, and hence the PF value, in two-dimensional electron gas thin-film applications. The technique involves a two-phase nanocomposite generated from two different types of nanograins. Figure 4a shows the operating concept of modulation doping’s generic design. Instead of the regular heavy uniform doping in TE materials, which results in significant ionised impurity scattering of charges, the modulation-doping scheme allows for even closer spacing of ionised nanoparticles, resulting in decreased electron scattering and increased mobility.

Likewise, Y. Pei et al. [81] observed a significant increase in TE efficiency in a BiCuSeO system through modulation doping, as shown in Figure 4b. They used a two-phase composite of undoped BiCuSeO and heavily doped Bi_0.75_Ba_0.25_CuSeO to increase the efficiency of their TE material. BiCuSeO exhibited a lower *n* but greater *μ*, while highly doped Bi_0.75_Ba_0.25_CuSeO exhibited a higher *n* but lower *μ*. The authors obtained high TE performance reflected by ZT = 1.4 at 923 K.

Another alternative is to use an ionised impurity scattering mechanism approach. Mao et al. [80] displayed that by doping transition metal elements such as Fe, Co, Hf, and Ta at the Mg site of MgSbBiTe, the prevailing ionised impurity scattering at lower temperatures changes to mixed acoustic phonons and ionised impurity scattering, thus significantly increasing the Hall mobility and PF. The results revealed that the fundamental concept of controlling carrier-scattering pro-cesses is based on defect control. The results revealed that the fundamental concept of controlling carrier-scattering processes is based on defect control. It is necessary to identify the defect that is responsible for the dominant carrier-scattering mechanism and then to try controlling it. This principle may apply to a variety of TE materials used to increase TE efficiency.

Another strategy is to utilise the carrier-scattering mechanism. Xiao et al. [79] reported TE material performance through introducing Cu_2_Te inclusions in *n*-type PbTe. The fact that Cu atoms are small enough to form interstitials in SnTe and efficiently scatter phonons encouraged them to explore the nature of Cu in PbTe. Beneficial effects of Cu doping are schematically shown in Figure 4c. Small amounts of Cu atoms fill the intrinsic Pb vacancies in PbTe. Eliminating Pb vacancies in this way diminishes carrier scattering and significantly improved *μ* by 5 times.

More recent work by Wang et al. [77] utilised the percolation effect to achieve nano-/microstructured BiSbSe_3_ composites by evenly combining the microscale and nanoscale grains. The nanostructured *n*-type BiSbSe_3_ sample prepared through mechanical alloying had dense grain boundaries that contributed to significant carrier scattering and disrupted *μ*. As can be seen in Figure 4d, the BiSbSe_3_ matrix composed of nanoscale and microscale grains could facilitate carrier transport and improve σ. This study suggested a convenient, effective method that optimised TE material quality by developing a microstructure with a percolation effect that could be applied to other structures.

Due to the fact that S and σ are related through *n* (electron carrier concentration), the optimal PF is typically obtained at a given *n*. Zhang et al. [86] succeeded in demonstrating that deep effect level can improve *n*. Previously, *n* has been chemically manipulated by the deliberate introduction of extrinsic dopants, which usually result in shallow levels of the materials. Such doping strategy will only optimise the TE performance in a limited temperature range. The authors employed indium to establish a deep-defect condition in PbTe, while iodine was used to provide enough electrons. At a lower temperature, the indium deep-defect states captured electrons, and the trapped electrons could be thermally reactivated to the conduction band at a higher temperature. This method effectively led to the optimisation of temperature-dependent *n* over a broad temperature range.

More recent work in this area by Kim et al. [83] extended the methods of enhancing *n* by using point-defect chemistry. They prepared Nb-doped Bi_2_O_2_Se through solid-state reactions and sintering to measure their TE parameter. The [Bi_2_O_2_]^2+^ layer is an insulating layer, whereas the [Se]^2-^ layer is a conducting layer with electron-conducting pathways. The [Bi_2_O_2_]^2+^ insulating layer causes Bi_2_O_2_Se to have low σ. The authors introduced Nb^5+^ dopants into Bi^3+^ sites to improve the σ. They reported that by point-defect chemistry, the partial substitution of Nb^5+^ ions for Bi^3+^ ions resulted in an increased *n*, enhancing the σ.
(8)$${\mathrm{Bi}}^{3+}{}_{{\mathrm{Bi}}^{3+}}\overset{{\mathrm{Nb}}^{5+}}{\to }{\mathrm{Nb}}^{5+}{}_{{\mathrm{Bi}}^{3+}}+2e^{-}$$

### 4.2. Advanced Approaches in Minimising Thermal Conductivity, κ

As mentioned in Equation (2), the ZT value is inversely proportional to κ. Therefore, materials with the possibly least κ are required. Thermal conductivities of solids vary with temperature and are determined by crystalline size variations, lattice defects or imperfections, dislocations, anharmonicity of lattice oscillations, carrier concentration, and interactions between carriers and lattice waves.

To reduce κ, researchers have developed various strategies, such as nanostructuring, lattice softening, complex crystal structuring, and rattling-like damping as a “resonant” phonon scattering. Hence, a summary of advanced approaches in minimising κ has been given. Different advanced approaches in reducing κ are shown in Table 4.

It has been mentioned previously that reducing${\;k}_{p}$ is an effective method to enhance ZT, since $k_{p}$ is the only independent parameter [105]. Among the methods to decrease $k_{p}$ is anharmonicity of the lattice oscillations. The Gruneisen parameter is sometimes referred to as a temperature-dependent anharmonicity parameter, since it indicates how often phonon vibrations deviate from harmonic oscillations in a crystal lattice. The phonon–phonon umklapp and standard processes that restrict k_p_ are controlled by the chemical bond’s anharmonicity [106]. A high Gruneisen parameter results in a lower κ. The high Gruneisen parameter may be the result of non-bonding valence electrons in the sp-hybridised bonding orbital [107]. H. Zhai et al. [90] reported ultralow k_p_ of 0.55 Wm^−1^K^−1^ in Cu_5_Sn_2_Se_7_ of its large Gruneisen parameter of 2 via aliovalent doping. Meanwhile, L. Zhao et al. [91] calculated the Gruneisen parameter using first-principles density functional theory (DFT) phonon calculations within the quasi-harmonic approximation to clarify the origin of the intrinsically low κ of their work on SnSe. The average Gruneisen parameter was found to be 4.1. They concluded that a high Gruneisen parameter is therefore a consequence of the ‘soft’ bonding in SnSe, which leads to the very low k_p_. Y.Pei et al. [92] substituted Bi^3+^ with Ca^2+^ in their work on *p*-type polycrystalline BiCuSeO. The Gruneisen parameter was determined to be 1.5, implying that the crystal lattices exhibit a strong degree of anharmonicity.

Nanostructuring can only operate where the phonon mean free paths are longer than the electron, such as when using sufficient grain size or introducing nanoinclusions of appropriate size to disperse phonons rather than electrons, according to theory. The first approach is to create coherent or semicoherent interfaces that allow carrier transfer while frequently hindering phonon propagation in the strained lattices. N. Wang et al. [96] employed yttria-stabilised zirconia (YSZ) nanoinclusion to reduce the k_p_ of SrTiO_3_. YSZ has a lower κ compared to SrTiO_3_. Figure 5a shows the effect of YSZ nanoinclusions inside the grain and in the triple junction. Nanoinclusion could scatter the phonons effectively and thus bring about the decrease in the phonon mean free path, and consequently lower the k_p_. Recently, Oh et al. [93] investigate nanomesh graphene and found a significantly low k_p_ compared to other suspended graphene nanostructures. As depicted in Figure 5b, the presence of circular edges in the nanomesh structure induces strong phonon scattering at these edge disorders.

R. Hanus et al. [98] focused on a PbTe model system that showed that the speed of sound linearly decreased with increased internal strain. The authors proposed that softening the lattice of the material completely accounted for the reduction in k_p_ without the introduction of additional phonon-scattering mechanisms. In this work, the internal strain was shown to be the result of Na-doped PbTe, leading to a high ZT value of 2. Internal strain fields locally changed the phonon frequency. Lattice softening refers to a decrease in the phonon speed, the magnitude of the group velocity vector (Figure 5c).

Another alternative to minimized k_p_ is to introduce filling guest atoms, functioning as ‘rattlers’ that are capable of inducing low-frequency vibration modes. These modes are strongly focused optical phonon modes with frequencies similar to those of acoustic modes. They are interrelated with the acoustic phonon modes and significantly constrain k_p_. Luo et al. [99] reported an ultralow κ via GeSe alloying for *n*-type PbSe. The theoretical studies revealed that alloyed Ge^2+^ atoms, functioning as rattlers preferred to stay at off-centre lattice positions, inducing very-low-frequency optical modes that caused softening of the phonon modes. More than one guest atom could be introduced as rattlers. Tan et al. [100] described low-frequency double-rattling modes for both Ag and Bi atoms in the AgBi_3_S_5_ compound. The authors performed a phonon-dispersion calculation and analysed the vibration behaviour, and found that the Ag and Bi atoms vibrated with much larger amplitudes than all other surrounding S atoms.

Low k_p_ may arise from materials with a complex crystal structure. Owing to the fact that the number of optical vibration modes increases linearly with the number of atoms (N) in the unit cell, a large number of optical phonons can occur in complex crystal structure compounds. Typically, optical phonons have a very limited group velocity, and contribute relatively little to thermal transport [108]. With increasing N, the increased amount of optical phonons continues to decrease the room and frequency of acoustic phonons, leading to the k_p_ decline [109]. Pei et al. reported that K_2_Bi_8_Se_13_ exhibited a complex low-symmetry crystal structure (Figure 5d) that led to an extremely low k_p_ of 0.20–0.42 Wm^−1^K^−1^.

### 4.3. Approaches for Improvements of n-Type Bi_2_Te_3_’s Poor Performance

In contrast to their *p*-type counterparts, ZTs of *n*-type polycrystalline Bi_2_(Te, Se)_3_ alloys are usually less than 1.0 due to the degraded texture and divergence from the carrier’s optimal concentration caused by ingot pulverisation and donor-like effects. The poor performance of *n*-type Bi_2_Te_3_ materials is a limitation that hinders a TE device from being more efficient [110]. A new technique with a simplified method must therefore be established to prepare superior *n*-type Bi_2_Te_3_ materials. Hence, we have provided a summary of the latest attempt by researchers worldwide to enhance the performance of *n*-type Bi_2_Te_3_.

X. Chen et al. [28] described a more straightforward and cost-effective process of ball milling and hot pressing than the time-consuming and expensive method of melting and spark plasma sintering. They successfully synthesised *n*-type Bi_2_Te_2.7+x_Se_0.3_ samples with an improved ZT peak that steadily moved upward as the excess Te content increased: a maximal ZT peak of 0.9 at 373 K, and an average ZT of 0.82 between 300–498 K. Greater antisite defects formed in a Te-rich atmosphere led to higher *n* and σ, a decreased$\;k_{p}$, and suppression of intrinsic excitation, which contributed to the enhancement of ZT at elevated temperatures. Thus, the increased ZT peak progressively moved to higher temperatures while the excess Te content was increased.

Y. Wu et al. [111] developed a liquid-phase hot-deformation (LPHD) technique in which the Bi_2_(Te, Se)_3_ ingots were directly hot-deformed through liquid eutectic phase extrusion. At 400 K, they observed a strong zT ≈ 1.1 in *n*-type LPHD Bi_2_(Te, Se)_3_ alloys. This study illustrated a straightforward technique for fabricating superior *n*-type Bi_2_Te_3_-based materials that were significantly more effective and energy-efficient than multiple hot deformations.

R. Zhai et al. [112] fabricated a material with a large ZT of 1.2 in Bi_2_Te_2.3_Se_0.69_. Along with hot-deformation processing, nonstoichiometry was used to mediate intrinsic point defects in *n*-type bismuth-telluride-based alloys. Highly pure element chunks of Bi, Te, and Se were weighted according to their nominal composition and sealed in a well-cleaned quartz tube at 10^−3^ Pa during the experiment. A Muffle furnace was used to heat the mixture in stages. The melts were rocked once every few hours to ensure that the Bi_2_Te_2.3_Se _0.7−x_ ingots were homogeneous. Through ball milling the ingot, fine powders were extracted. The powders were hot-pressed for 30 min. Immediately afterward, a hot-deformation procedure was used to compress the bulk into a 20 mm disc at 823 K for 30 min. They concluded that when Bi_2_Te_2.3_Se_0.69_ was subjected to one-time hot deformation, the *n* increased due to the induced donor-like influence, while improving the texture and boosting the PF. Proper Se deficiency resulted in enough point defects to efficiently scatter phonons, resulting in a decreased k_p_.

## 5. TE Materials at Low Operating Temperatures

The greatest need for new TE materials is at low operating temperatures. TE operating in a low-temperature area is highly needed for recycling low-grade waste heat. According to [113], for TE working at near-room temperature, for the material required to have a combination of high ZT, low κ, and high S, unlike TE working at intermediate or high temperature, the low value for κ is not essential. Thus, there is a motivation to improve the material properties to enhance the TE performance in this temperature region.

Currently, there is no bulk material with a high enough TE efficiency to reach a practical level in the low-temperature region below 250 K. As a result, there is a drive to upgrade material properties in an attempt to optimise TE efficiency in this temperature range. To achieve such high efficiency, improving the PF and S = σS^2^ of a material, where S and σ are the TE power and σ, respectively, is the first step. A target value of PF is ~35 μWcm^−1^K^−2^, which is the typical value of Bi_2_Te_3_-based practical materials [60]. Several candidate materials working at low operating temperatures, such as Bi_2_Te_3_, CsBi_4_Te_6_, AgSbTe_2_, and MgAgSb, have been developed thus far. The basic physical parameter of near-room temperature TE and strategies that have been taken to optimise them are explained in the following sections.

### 5.1. Bismuth Telluride, Bi_2_Te_3_

At present, bismuth–telluride-based alloys are the most efficient TE material working at room temperature [40,114,115]. The high S and low κ are the primary reason for this [116,117,118]. Based on works done in [119], the maximum ZT value for Bi_2_Te_3_-based TE materials is obtained only at room temperature, and with increasing temperature, ZT sharply drops. Thus, Bi–Te-based alloys are suitable for working at around room temperature.

Bi_2_Te_3_ forms single crystals that are markedly anisotropic in their mechanical properties. Bi_2_Te_3_ crystal is layered in hexagonal symmetry. The layers are repetitively stacked in a five-layer sequence (Te–Bi–Te–Bi–Te), so two Te layers are adjacent on the boundaries of these units [60]. On both forms of sites, strong covalent–ionic interactions occur between the Bi atoms and the Te atoms, but the layers of Te atoms are only held together by weak van der Waals forces. Bi_2_Te_3_ crystals can be conveniently cleaved along the layer path, which is common to the trigonal or c-direction. Two a-axes are perpendicular to the c-axis and are at a 60° angle to each other. It is not just the mechanical properties that vary between the a-axes and the c-direction planes. For instance, parallel to the cleavage planes, electrical and thermal conductivities are greater than when perpendicular. The k_p_ of single-crystalline Bi_2_Te_3_ was experimentally determined as 1.5 W/(m·K) at 300 K along the basal plane (perpendicular to the c-axis) and 0.7 W/(m·K) along the lateral plane (parallel to the c-axis) [120].

The TE properties of Bi_2_Te_3_ materials are changed by different doping elements. Conventional doping is usually done with elements Sb and Se. Excess Bi atoms appear to act as acceptor impurities, leading to *p*-type conduction. Bi-rich Bi_2_Te_3_ is *p*-type, owing to defects in the BiTe antisite acceptor. Bi substitutes Te with an electrical configuration that is identical to Te but has one less electron, resulting in a void in the valence band. On the other hand, Sb_2_Te_3_ exhibits SbTe_1_ antisite defects, indicating a general *p*-type tendency with the lowest formation energy across a wide range of growth conditions, most notably in the Sb-rich condition. When the growth surroundings become extremely Te-rich, the most energetically stable vacancy is the antimony vacancy, V_Sb_. Notably, both SbTe_1_ and VSb are acceptor-like defects; hence, Sb_2_Te_3_ is indeed intrinsically *p*-type [121]. Also, *n*-type conductivity can be observed in Bi_2_Se_3_ due to donor-like defects, V_Se1_ and Se_Bi_. Indeed, the formation of compounds is facilitated by their low formation energies, which vary according to the growth conditions [122].

Due to the high mean atomic weight of Bi_2_Te_3_, it was chosen as a low-TE material. Moreover, it has a comparatively low melting point of 585 °C [62]. The fact that Bi_2_Te_3_ possessed higher weighted mobility for electrons (μW = 525 cm^2^·V^−1^·s^−1^) compared to holes (400 cm^2^·V^−1^·s^−1^) gives a larger PF for *n*-type materials (≈45 μW·cm^−1^·K^−2^), while *p*-type samples reach only ≈35 μW cm^−1^·K^−2^. At 300 K, all *n*- and *p*-type charge carriers exhibit a comparable Seebeck $m^{*}$ of 1.06 m_e_. By extracting the electronic input to κ, we obtain an average k_p_ of 1.37 W m^−1^·K^−1^ (300 K) for single-crystal Bi_2_Te_3_ [123]. ZT at 300 K has been reported to have a maximum value of 1.0 for coarse-grained *p*-type (Bi-Sb)_2_Te_3_, and a significantly higher value for nanostructured material. ZT is nearly 0.9 for aligned *n*-type Bi_2_(Se-Te)_3_, although a ZT value closer to 0.6 is more common for randomly directed polycrystalline samples [114]. For industrial modules, the average ZT value lies between 0.5 and 0.8 [124], leading to the TE device’s low thermal efficiency (less than 4%). Table 5 summarises the ZT value of different Bi–Te based material.

Scientists worldwide have performed comprehensive research in order to increase the TE efficiency of materials based on Bi_2_Te_3_ and to extend the application of electricity generation. The goal is to synthesise high-performance materials based on Bi_2_Te_3_ with various methods and hypotheses so that their transport properties are improved and the ZT value is substantially increased. Bulk TE materials are mainly centred on the semiclassical electron-transport theory to measure ZT’s TE efficiency, and the bulk TE materials are typically low in ZT. Thus, the low-dimensional nanometer is a critical aspect affecting the quality of TE materials. In recent years, there has been a surge in interest in research on Bi_2_Te_3_-based TE nanostructures, especially superlattice structures and nanowires [132]. The Bi_2_Te_3_-based TE has reportedly been developed by various methods such as high-energy ball milling (HEBM), hot pressing (HP), melt spinning (MS), spark plasma sintering (SPS), hydrothermal synthesis (HS), and zone melting. Table 6 displays the TE properties of nanostructured materials synthesised using different synthetic methods.

### 5.2. Cesium Tetrabismuth Hexatelluride, CsBi_4_Te_6_

CsBi_4_Te_6_ as a TE material was first reported by A. Chung et al. [143]. They described the newly found compound, in the form of shiny and long silvery needles, (Figure 6a) as a seemingly outstanding candidate for low-temperature TE applications, as shown in Figure 6b. The properties of this material can be seen in Table 7.

Furthermore, Chung et al. found that CsBi_4_Te_6_ crystallises in the space group C2/m and has a layered anisotropic structure. It is made up of anionic [Bi_4_Te_6_]^−^ slabs and Cs^+^ ions in the interlayer region. CsBi_4_Te_6_ is a reduced version of Bi_2_Te_3_; moreover, the addition of one electron from Cs per two equivalents of Bi_2_Te_3_ does not result in a formal intercalation compound. However, the additional electrons trigger a radical transformation of the Bi_2_Te_3_ structure and localization on a Bi atom, resulting in formally Bi^2+^ (rare in Bi chemistry) and Bi–Bi bonds of 3.2383(10) Å.

The band structure for CsBi_4_Te_6_ is in an ultranarrow gap of 0.04 eV. This value corresponds to the energy-gap spectrum of 0.04–0.08 eV derived from the formula Eg ≈ 2S_max_T_max_, where S_max_ denotes the maximum S, and T_max_ denotes the temperature at maximum S. W. Luo et al. [157] calculated the electronic structure of CsBi_4_Te_6_ by means of first-principle, self-consistent total-energy calculations within the local density approximation. They successfully calculated the frequency-dependent dielectric function, which indicated that CsBi_4_Te_6_ is a semiconductor with a 0.3 eV bandgap. The calculated density of state supports the experiment of Chung et al. [148], confirming that CsBi_4_Te_6_ is a high-performance TE material for low-temperature applications. Initially, five CsBi_4_Te_6_ synthesis methods were documented according to [148]. Recently, multiple experiments have been published related to the new path of CsBi4Te6 synthesis.

Recently, H. Lin et al. [147] reported a new simple approach to CsBi_4_Te_6_ synthesis method by not using the reactive Cs metal. They deliberated a synthetic process for CsBi_4_Te_6_ utilising rare earth metals as the reducing reagents, which had not been attempted. The major advantage of this route was the safety of not using the reactive Cs metal. The compound was prepared from a mixture of praseodymium (Pr), Bi, and Te, together with CsCl flux at 1173 K, by solid-state reactions. Such a process generated CsBi_4_Te_6_ ingot inside the silica. The obtained ingot was first washed with distilled water to remove excess flux and chloride byproducts, and then dried with ethanol.

In contrast, N. Gostkowska et al. [146] synthesised CsBi_4_Te_6_ polycrystalline materials using a cost-effective approach focused on oxide-reduction reagents. The authors proposed a method for TE material fabrication using the reduction of melted oxides. Cs_x_Bi_4_Te_6_O_y_ (where x = ¼, 1, and 1.1) samples were made from a sufficient volume of oxides and cesium carbonate. The powders were then combined to produce a stoichiometric sample (CsBi_4_Te_6_-melted oxides) and a surplus cesium sample (Cs_1.1_Bi_4_Te_6_-melted oxides). The powders were dissolved at 1050 °C and then air-cooled. The oxides were in a liquid state at 1050 °C. The collected polycrystalline content was ground and then uniaxially pressed at 350 MPa. Bulk cylindrical pellets were obtained and reduced in hydrogen at 400 °C for 10 h. Following the preliminary reduction, pellets were ground and pressed with a pressure of 700 MPa. Finally, the samples were cooled to 400 °C for 10 hours in hydrogen.

### 5.3. Silver Antimony Telluride, AgSbTe_2_

Silver antimony telluride, or AgSbTe_2_, is a crystalline material with a low κ and higher TE efficiency than lead telluride; its nanocrystalline structure has been studied by researchers as a potential model for new materials for automotive and energy-production technologies. The AgSbTe_2_ compound has been repeatedly studied as a prospective *p*-type TE material for use in the temperature range T = 400–700 K [158].

The complexity of ordering of Ag/Sb on the face-centred lattice and strong lattice vibrational anharmonicity contributes to the large S and low κ of this material [159]. Ag_2_Te is a narrow-gap semiconductor that operates at low temperatures (α-Ag_2_Te, Eg = 0.064 eV). It is a nominal rock-salt (Fm$\bar{3}$m space group) narrow-band-gap semiconductor, with Ag^+^ and Sb^3+^ sharing the cation sublattice. When heated to 423 K, Ag_2_Te undergoes a structural phase change from low-temperature monoclinic α-Ag_2_Te to high-temperature cubic phase β-Ag_2_Te, resulting in superionic conductivity, since Ag cations can easily pass through the cubic sublattice created with Te atoms. Among the ternary chalcogenides, AgSbTe_2_ exhibits the highest ZT value of about 1.3 at 720 K, consistent with its large S of about 200 μV/K and low k_p_ of about 0.6–0.7 W/mK [160].

Early reports revealed that samples were obtained by melting and slow-cooling [158], melt spinning [151], mechanical alloying [161], sonochemical method [162], and high-pressure and high-temperature techniques [163]. Recently, T. Zhu et al. [144] synthesised high TE efficiency in Sb-doped Ag_2_Te compounds with a monoclinic structure at low temperatures. They recorded a 120% increase in performance over pure Ag_2_Te, as shown in Figure 6c. The vacuum melting and annealing method was used to synthesise a sequence of samples with the nominal composition Ag_2_Sb_x_Te_1-x_. Stoichiometric quantities were measured and mixed for Ag strips, Te bulks, and Sb grains. The compositions were sealed into evacuated quartz tubes and put in an oven heated at 1.5 K/min to 1273 K, and the temperature was maintained for 10 h. During the melting phase, the quartz tube was rocked at 1273 K every 2 h to reach homogeneous melts. The tubes were then eventually cooled down to 673 K at 1 K/min. The resulting ingots were then annealed at 673 K for 120 h before being cooled to 100 K and kept for 48 h. The obtained ingot was cut into the required shape for characterisation and performance evaluation. The properties of this material can be seen in Table 7.

### 5.4. Magnesium Silver Antimonide, MgAgSb

MgAgSb material has been attracting more attention as a highly efficient TE material near room temperature for its nontoxicity and elemental abundance of its raw materials. As initially documented by Kirkham et al. [164], MgAgSb in its pure state undergoes complex step transformations, from ambient temperature to intermediate temperature and eventually to high temperature. Unlike α-MgAgSb, these two phases (β and γ) have worse TE properties. The primary distinction between the three MgAgSb structures is in the pattern of the Ag atom filling the Mg–Sb cubes. α-MgAgSb is a distorted Mg–Sb rock-salt sublattice that rotates 45° around the c-axis, as shown in Figure 6d. If the temperature increases, a trend toward increased structural symmetry may be observed. The properties of this material can be seen in Table 7.

Pure MgAgSb phase synthesis is so challenging that Kirkham et al. initially did not obtain phase pure samples using the conventional synthesis method. Their specimens contained a large concentration of secondary phases, specifically Sb and Ag_3_Sb. Furthermore, the composite sample showed a maximum zT of only ~0.5 at 425 K. Then, Zhao et al. used an ordinary two-step ball milling and hot-pressing method to fabricate MgAgSb compounds, and obtained improved ZT [155]. They found that the grains in the samples made by ball milling and hot pressing were smaller than 20 nm, which was much smaller than those in other TE materials made by the same method. However, ball-milled Mg-based alloys are prone to be oxidised. Ying et al. [156] successfully obtained high-purity α-MgAgSb by elaborately controlling the SPS processing. Zhen et al. [154] reported that heat-treating for 10 days had proven to strengthen the transformation of β-MgAgSb to α-MgAgSb and its consumption of Sb and Ag_3_Sb, resulting in higher purity (α-MgAgSb) of the samples. Recently, Li et al. [153] utilised the microwave-assisted process and spark plasma sintering to fabricate MgAgSb samples. The maximum PF was nearly twice larger than that of the sample prepared by the melting process. The maximum ZT value, however, was less than 1, implying considerable areas for growth.

## 6. Conclusions and Future Research

In recent years, the efficiency of TE generator devices has been improved significantly thanks to improvements in materials and better knowledge of the working mechanisms of the devices. According to previous reports, the global exploratory study of TE materials has increased significantly, to the point that the previously predicted maximum limit of ZT ≈ 1.0 has been exceeded through the implementation of novel design approaches and materials. The performance of TE materials can be enhanced through either optimizing power factor (PF) properties or minimizing the thermal conductivity, κ. The performance of TE materials can also be enhanced through the synergistic effects of these properties.

In this review, the current progress of TE materials at low operating temperatures were reported. Numerous candidate materials for low-operating-temperature applications have been proposed thus far, including Bi_2_Te_3_, CsBi_4_Te_6_, AgSbTe_2_, and MgAgSb. The fundamental physical parameters of near-room temperature TE materials and the methodologies used to optimise them were described. TEG technology at low operating temperatures is projected to allow for renewable power production, and is the main technology for large-scale promotion with wide potential applications; an abundant source of waste heat that provides benefits for global climate change; has no mechanical moving parts; and offers stability, no noise, and flexible incorporation. However, the realistic implementation of TEG technology has several obstacles. The limitations and corresponding research directions are proposed as follows:
(1)Improvement of TE efficiency, especially for *n*-type TE materials, is the most critical issue at the moment. The ZT performance of TE products has dramatically changed in recent years thanks to phonon engineering and energy-band strategies. However, the ZT value for low-operating-temperature TE material is still around unity, causing the low efficiency of TE systems. Further advancements would necessitate a significant amount of creativity in order to investigate different mechanisms and methods for developing TE properties.(2)High-performance TE materials at room temperature make use of low-abundance components found in the Earth’s crust, such as Bi_2_Te_3_. However, the widespread use of Bi_2_Te_3_-based TE is limited by the low abundance, relatively high cost, and toxicity of the Te element. There is still an urgent need to develop alternative green TE materials with abundantly available elements in the crust and good TE properties near room temperature.(3)Research on potential near-room temperature TE materials needs to be expanded. At the moment, the in-depth analysis focuses primarily on standard Bi_2_Te_3_-based TE. Research on other TE materials based on CsBi_4_Te_6_, AgSbTe_2_, and MgAgSb is still limited. In addition, the potential candidates for low-operating-temperature TE materials are only laboratory technologies at present, and large-scale, low-cost preparation technologies need to be explored further.(4)Very recently, research on graphene, single-walled carbon nanotubes (SWCNTs), and TE material nanocomposites demonstrated better ZT performance, as well as improved electrical conductivity (σ) and were successful in suppressing the thermal conductivity, κ [27,165,166,167]. The κ of TE nanocomposites is predicted to be lower than that of their bulk equivalents of the same chemical configuration. They can be manufactured at a low price by combining nanoparticles and nanosized powders.


All in all, we expect that, based on available data, material efficiency will hit an average ZT value of 2 over the next 10 years, at which point TE technology will begin to be used on a massive scale in low- and medium-temperature waste-heat energy-harvester systems.

## Figures and Tables

**Figure 1 micromachines-12-00734-f001:**
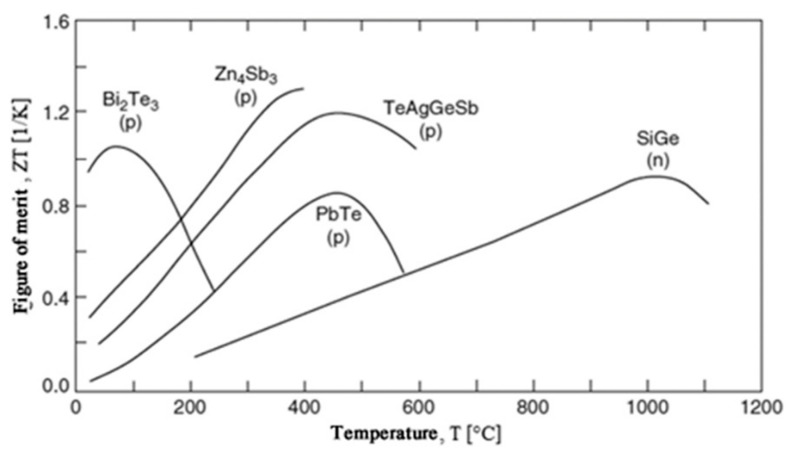
ZTs of common TE materials for *p*- and *n*-types. Reprinted with permission from [41].

**Figure 2 micromachines-12-00734-f002:**
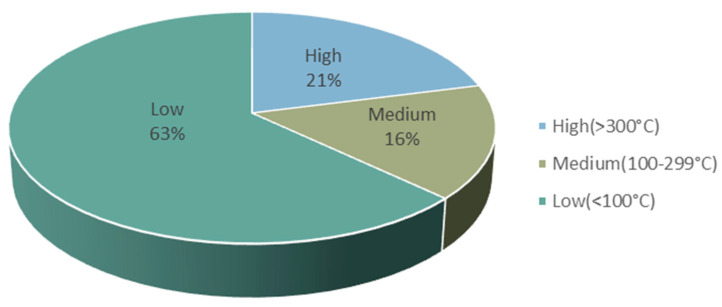
Waste-heat temperature distribution. Adapted from [57].

**Figure 3 micromachines-12-00734-f003:**
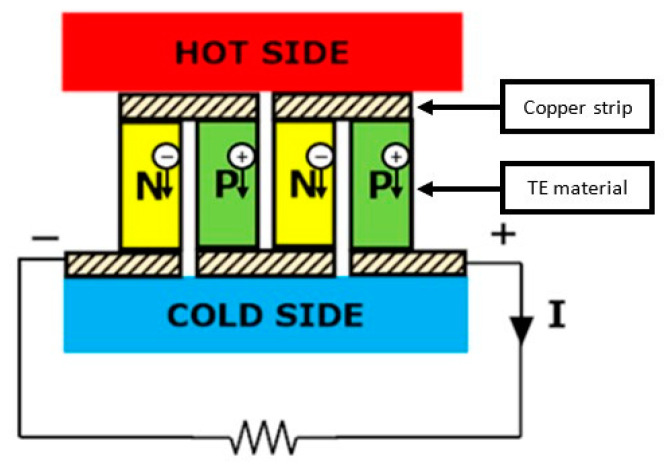
The *p*- and *n*-type TE materials placed between the cold side and the hot side.

**Figure 4 micromachines-12-00734-f004:**
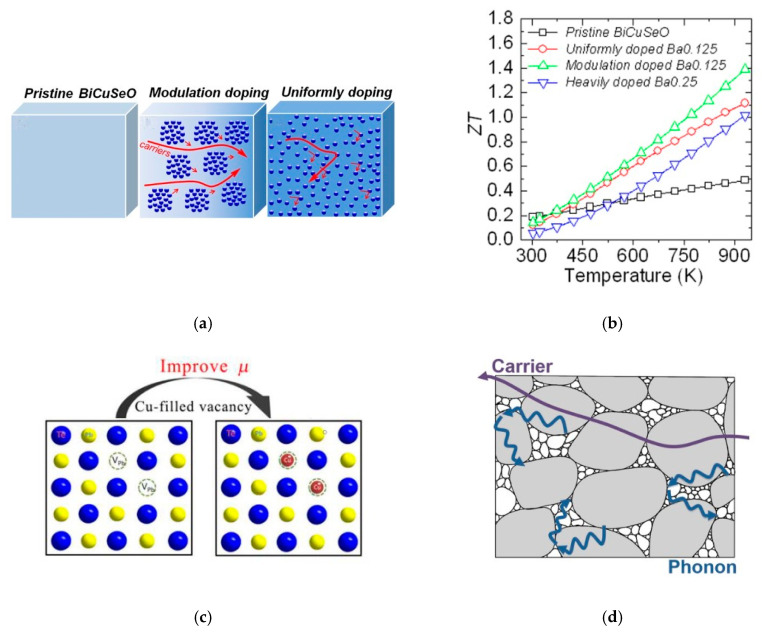
(**a**) Schematic representations of pristine BiCuSeO, modulation-doped sample, and heavily doped sample. Reprinted with permission from [81]. (**b**) ZT as a function of temperature for the pristine, modulation-doped, uniformly doped, and heavily doped BiCuSeO. Reprinted with permission from [81]. (**c**) An illustration of intrinsic Pb vacancies filled by Cu atoms to improve *μ*. Reprinted with permission from [79]. (**d**) Schematic representations of carrier and phonon-transport paths in the composite material with nanoscale and microscale grains. Reprinted with permission from [77].

**Figure 5 micromachines-12-00734-f005:**
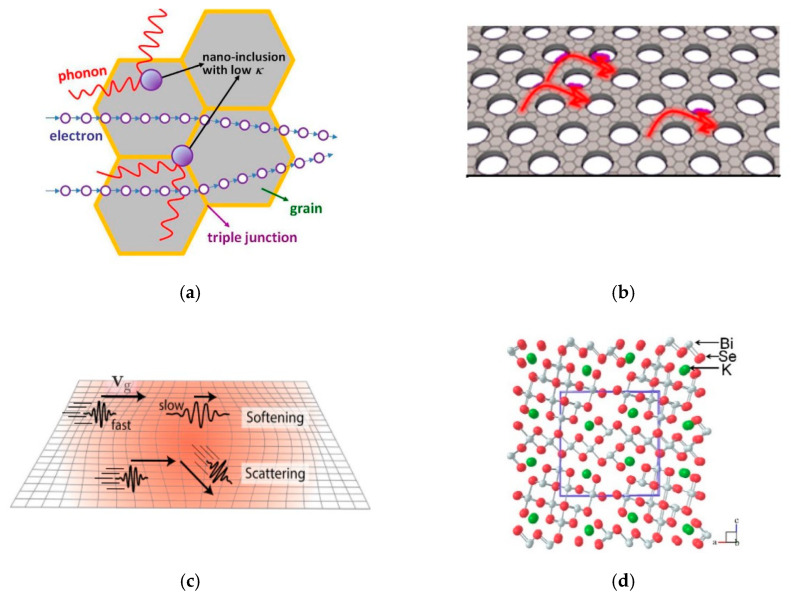
(**a**) Schematic representations of phonon-scattering effects due to nanoinclusion. Reprinted with permission from [96]. (**b**) Schematic representations of transport behaviour of phonons controlled by nanomesh-provided edge scattering. Reprinted with permission from [93]. (**c**) Schematic representations of phonon-scattering and lattice-softening effects due to internal-strain fields. Reprinted with permission from [98]. (**d**) Complex low-symmetry crystal structure of K_2_Bi_8_Se_13_. Reprinted with permission from [104].

**Figure 6 micromachines-12-00734-f006:**
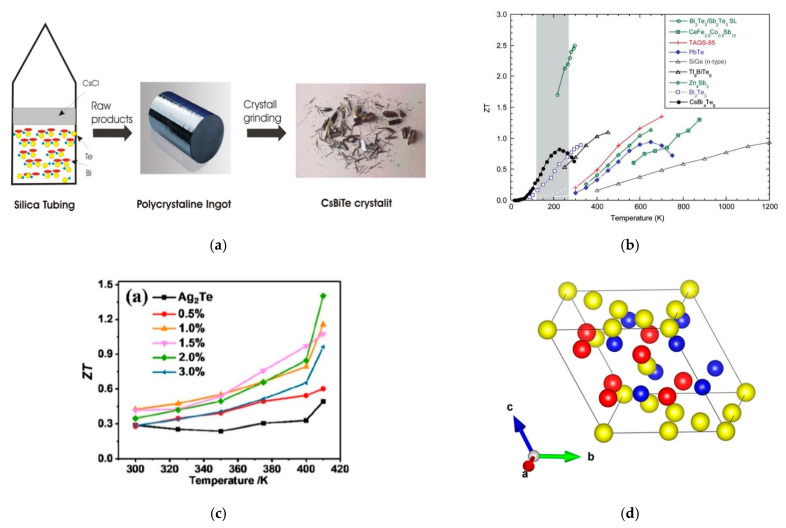
(**a**) The schematic synthesis process of CsBi_4_Te_6_ crystals [143]. (**b**) ZT of thermoelectric materials as a function of temperature. In the shaded region, CsBi_4_Te_6_ could possibly be available for thermoelectric applications. Reprinted with permission from [143]. (**c**) ZT value as a function of temperature for Ag_2_Sb_x_Te_1−x_ samples. Reprinted with permission from [144]. (**d**) The crystal structure of α-MgAgSb. Reprinted with permission from [145].

**Table 1 micromachines-12-00734-t001:** Different waste-heat sources and temperature ranges for TEG.

Temperature Ranges, K	TE Material	Temperature, K	Example of Waste-Heat Sources
Low temperature (around 400)	Bi_2_Te_3_ alloy [25,26,27,28,29,30]	305–330 300–323 300–360 310 <360 350–400	Cooling water Air compressors Forming dies and pumps Body heat Low-temperature geothermal Automotive engines
Medium temperature (600–900)	PbTe alloy [31,32] TiO_2_ [33,34] Skutterudite [35]	315–600 425–650 425–650	Engine exhausts Catalytic crackers Furnace
High temperature (>900)	Half-Heusler [36,37] SiGe [38,39]	650–760 760–1100 620–730	Aluminium-refinement furnace Copper-refinement furnace Hydrogen plants

**Table 2 micromachines-12-00734-t002:** Potential applications of low-operating-temperature TEGs.

Application	Powered Devices	Working Temperature, K	References
Implantable biomedical system	Electrocardiographic systems Oximeter	Body temperature	[43,44,45]
Wearable electronics	E-skin sensors, temperature sensors	Body temperature	[46,47,48,49]
Environmental monitoring	Humidity sensors, temperature sensors, wireless sensor nodes	300–420	[50,51]
Building	Smart window	100–400	[33,52]
Automotive	E-vehicle	370–400	[53,54,55]

**Table 3 micromachines-12-00734-t003:** Different advanced approaches in enhancing PF, clustered based on the enhanced physical properties ($m^{*}$, *n*, and *µ*).

Authors	Published Year	Material	Enhanced Physical Property	Enhanced PF Value (μWcm^−1^·K^−2^)	ZT	Method
Shen et al. [71]	2020	BiCuSe	$m^{*}$	6.8–7.1	0.29	Band engineering
Du et al. [72]	2020	GeSbTe	$m^{*}$	2.1–4.2	0.41	Convergence of band gap
He et al. [73]	2019	SnS_0.91_Se_0.09_	$m^{*}$	30–53	1.6	Convergance of electronic band
Tan et al. [74]	2019	PbTe	$m^{*}$	5.6–17	1.2	Band engineering
Xiao et al. [75]	2018	PbTe	$m^{*}$	19.7–23.7	1.6	Flatten conduction band
Althaf et al. [76]	2020	ZnO	*µ*	1.5–15.2	-	Grain-boundary engineering
Wang et al. [77]	2020	BiSb	*µ*	4.6–6.0	1.0	Percolation effect
Feng et al. [78]	2018	BiCuSeO	*µ*	2.8–7.2	1.17	Modulation doping
Xiao et al. [79]	2017	PbTe	*µ*	21.9–36.7	1.5	Carrier-scattering mechanism
Mao et al. [80]	2017	Mg_3_Sb_2_	*µ*	5–13	1.7	Ionized-impurity scattering
Pei et al. [81]	2014	BiCuSeO	*µ*	5–10	1.4	Modulation doping
Yu et al. [82]	2012	SiGe	*µ*	-	1.3	Modulation doping
Kim et al. [83]	2021	Bi_2_O_2_Se	*n*	2.7–7.0	-	Point-defect chemistry
W.Liu et al. [84]	2020	GeTe	*n*	-	>2	Band engineering
M.Hong et al. [85]	2020	GeTe	*n*	7.2–11.6	2.2	Band convergence
Zhang et al. [86]	2018	PbTe	*n*	14–26	1.4	Deep-defect level
Berry et al. [87]	2017	TiNiSn	*n*	30–45	0.63	Modulation doping
Zhang et al. [88]	2015	PbSe at	*n*	-	1.0	Carrier engineering
Pei et al. [89]	2011	PbTe	*n*	-	1.7	Band engineering

**Table 4 micromachines-12-00734-t004:** Different advanced approaches on minimising κ.

Authors	Published Year	Material	Thermal Conductivity W/mK	ZT	Method
Zhai et al. [90]	2020	Cu_5_Sn_2_Se_7_	2.49–0.55	0.51	Anharmonicity
Zhao et al. [91]	2014	SnSe	0.68–0.23	2.6	Anharmonicity
Pei et al. [92]	2013	BiCuSeO	0.89–0.45	0.9	Anharmonicity
Oh et al. [93]	2017	Graphene nanomesh	3000–78	-	Nanostructuring
Lee et al. [94]	2016	SnS_2_	10.0–3.0	0.13	Nanostructuring
Rahal et al. [95]	2016	SnTe	8.1–5.3	1.35	Nanostructuring
Wang et al. [96]	2013	SrTiO_3_	15% reduction	0.21	Nanostructuring
Banik et al. [97]	2019	SnTe	2.89–0.67	1.6	Local structural distortions
Hanus et al. [98]	2019	PbTe	20% reduction	2.0	Lattice softening
Luo et al. [99]	2018	PbSe	4.69–1.52	1.54	Incoherent rattling motion
Tan et al. [100]	2017	AgBi_3_S_5_	0.6–0.3	1.0	Double-rattling behaviour
Zhen et al. [101]	2018	GeTe	3.37–0.13	1.61	Alloy scattering
Yang et al. [102]	2018	BaCu_2_Te_2_	0.9–0.5	1.08	Point defects
Samanta et al. [103]	2017	GeTe	3.0–0.7	2.1	Point defects
Pei et al. [104]	2016	K_2_Bi_8_Se_13_	0.5–0.2	1.83	Complex crystal structure

**Table 5 micromachines-12-00734-t005:** The ZT values of Bi–Te-based materials.

Authors	Material	ZT	Temperature
Amin et al. [113]	Bi_2_Te_2.7_Se_0.3_	0.87	Room temperature
Ju et al. [118]	Graphene/Bi_2_Te_3_-NW	0.4	Room temperature
Fadzli et al. [125]	Pt/Bi_2_Te_3_	0.61	Room temperature
Tan et al. [126]	Bi_2_Se_0.5_Te_2.5_	1.28	Room temperature
Yeo et al. [127]	(Bi,Sb)_2_Te_3_	1.41	Room temperature
Tan et al. [128]	Bi_2_Te_2.7_ Se_0.3_	1.27	Room temperature
Chen et al. [129]	Bi_0.4_Sb_1.6_Te_3_	1.26	Room temperature
Tan et al. [130]	Bi_2_(Te, Se)_3_	1.01	Room temperature
Xu et al. [131]	*p* type(Bi_0.26_ Sb _0.74_)_2_Te_3_ + 3%Te ingots	1.12	Room temperature

**Table 6 micromachines-12-00734-t006:** TE properties of nanostructured materials by various synthetic methods.

Material System	Carrier Type	ZT	T (K)	Synthetic Method
(Bi,Sb)_2_Te_3_ [133]	*P*	1.5	390	MS + SPS
(BiSb)_2_Te_3_ [134]	*P*	1.47	440	HS + HP
Bi_0.52_Sb_1.48_Te_3_ [135]	*P*	1.56	300	MS + SPS
Bi_2_Te_2.7_Se_0.3_ [136]	*N*	1.04	498	HEBM + HP
Bi_2_Te_3_ [137]	*N*	1	450	HS + HP
Bi_2_(Te_1-x_S_ex_)_3_ [138]	*N*	0.8	600	Zone melting
Cu_0.01_Bi_2_Te_2.7_Se_0.3_ [139]	*N*	1.06	-	Nanostructuring
Bi_2_Te_2.7_Se_0.3_ [140]	*N*	1.23	480	Nanostructuring
Bi_2_Te_2.7_Se_0.3_(AZO)_0.005_ [141]	*N*	0.85	323	Nano inclusions
Bi_2_Te_3_ [142]	*N*	1	513	Texturing

**Table 7 micromachines-12-00734-t007:** Properties of some other potential TE materials.

Material	S (*μ* V.K^−1^)	k (Wm^−1^K^−1^)	ZT	Method
CsBi_4_Te_6_ [143]	90	1.85	0.8	
CsBi_4_Te_6_ [146]	−70	0.43	0.054	Reduction of oxide reagents
CsBi_4_Te_6_ [147]	76	1.07	0.14	Rare-earth metals
CsBi_4_Te_6_ [148]	−74	1.25	0.82	
Ag_2_Sb_x_Te_1−x_ [144]	−103	0.35	1.4	Sb doping
AgSbTe_2_ [149]	58	-	0.74	Mn doping
SnTe-AgSbTe_2_ [150]	160	0.4	1.2	I doping
Ag_2_Te-Sb_2_Te_3_ [151]	300		1.4	Se doping
AgSbSe_x_Te_2−x_ [152]		0.6	1.37	Se doping
α-MgAgSb [153]	189	1.06	0.76	Microwave method
MgAgSb [154]	278	1.15	1.4	Heat-treating
MgAgSb [155]	210	0.8	0.9	Ball milling
MgAgSb [156]	190	1.10	1.1	In doping

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
