# Peer review of "Review of Thermoelectric Generators at Low Operating Temperatures: Working Principles and Materials"

_micromachines, 2021, doi:10.3390/mi12070734_

Round 1
Reviewer 1 Report
This article is summarized the fundamental working principle of the TEG and the recent TE materials that available at room temperature as well as the advanced method to optimize their ZT value. This work can support relevant research and provide a potential research direction on the materials of TEG at room temperature. However, there are still some suggestions for this article.
1. In my opinion, from the title of the article, this article is not completely for the materials of TEG, the relevant devices are also suggested to be provided to make the article become more convinced.
2. The approaches for improving the performance of n-type Bi2Te3 is suggested to summary in section 4 rather than in section 5, which can make the structure of the article is more logical. The reason is the main approaches for improving the performance of materials is in section 4 not in section 5.
3. The multiple schematic images are suggested to summary in one image. For example, Figure 6-9 are all talking about Phonon transport and scattering, which can be integrated into one image.
4. The relevant schematic illustration of the materials in section 5 should be provided. For example, a schematic illustration about the synthetic process of CsBi4Te6 and its performance in section 5.2, which can make the article become more attractive and convinced.
5. The potential applications of thermoelectrics and their generators should also be summarized, such as thermoelectrics (Chem. Eng. J. 2020, 397, 125360; Renew. Sust. Energ. Rev. 2021, 141, 110800; Adv. Energy Mater. 2020, 10, 2000367; Adv. Sci. 2020, 2001362;Energy & Environmental Science 14 (2), 729-764; InfoMat 2020, 2 (6), 1201-1215; Joule 2020, 4 (9), 2030-2043; Energy & Environmental Science 2020, 13 (6), 1856-1864), where these publications are also attracting extensive attentions.
Reviewer 2 Report
This is a very nice overview paper, however, some work still has to be done.
- I think that the paper should be structured more clearly. I am aware that this a review paper, on the other had, even here the section on the Methodology should be inserted (as independent section) so that we could clearly see and understand the flow of the work (maybe even graphical scheme would be very informative here).
- I also think that the section where the original results are articulated, should be stronger.
- Please check the paper for typos and grammatical errors.
- Please check the format of references as it seems to me that these are not in line with the Guide (first name of authors should not be used).
- Please check the abbreviations throughout the manuscript if all are correctly explained.
- I would proposed to make the abstract easier reading so that more reader can be attracted. Please use a layman style here.
- In the Conclusion, please make stronger the part where the findings are presented. I also think that the limitations of the approach used in the paper should be commented here.
Round 2
Reviewer 1 Report
I suggest to publish this work
Reviewer 2 Report
From my perspective, the manuscript can be accepted for publication as the vast majority of my comments and observations was reflected or replied.
Let me congratulate the authors for the new paper.
Kind regards,